# A Functional Virgin Olive Oil Enriched with Olive Oil and Thyme Phenolic Compounds Improves the Expression of Cholesterol Efflux-Related Genes: A Randomized, Crossover, Controlled Trial

**DOI:** 10.3390/nu11081732

**Published:** 2019-07-26

**Authors:** Marta Farràs, Sara Arranz, Sílvia Carrión, Isaac Subirana, Daniel Muñoz-Aguayo, Gemma Blanchart, Marjon Kool, Rosa Solà, María José Motilva, Joan Carles Escolà-Gil, Laura Rubió, Sara Fernández-Castillejo, Anna Pedret, Ramón Estruch, María Isabel Covas, Montserrat Fitó, Álvaro Hernáez, Olga Castañer

**Affiliations:** 1Molecular Bases of Cardiovascular Risk Group, IIB-Sant Pau, 08041 Barcelona, Spain; 2CIBER de Diabetes y Enfermedades Metabólicas Asociadas (CIBERDEM), ISCIII, 28029 Madrid, Spain; 3Cardiovascular Risk and Nutrition Research Group, Hospital del Mar Medical Research Institute (IMIM), 08003 Barcelona, Spain; 4CIBER de Epidemiología y Salud Pública (CIBERESP), Instituto de Salud Carlos III (ISCIII), 28029 Madrid, Spain; 5Cardiovascular Genetics and Epidemiology Research Group, IMIM, 08003 Barcelona, Spain; 6CIBER de Fisiopatología de la Obesidad y Nutrición (CIBEROBN), ISCIII, 28029 Madrid, Spain; 7Functional Nutrition, Oxidation, and Cardiovascular Diseases Group, Universitat Rovira i Virgili, 43201 Reus, Spain; 8Institut d’Investigació Sanitària Pere Virgili, 43204 Reus, Spain; 9Hospital Universitari Sant Joan de Reus, 43204 Reus, Spain; 10Instituto de Ciencias de la Vid y el Vino-ICVV (Consejo Superior de Investigaciones Científicas-CSIC, Gobierno de La Rioja, Universidad de la Rioja), 26007 Logroño, Spain; 11Food Technology Department, Agrotecnio Center, University of Lleida, 25198 Lleida, Spain; 12Internal Medicine Service, Hospital Clínic de Barcelona, 08036 Barcelona, Spain; 13Cardiovascular Risk, Nutrition and Aging Research Unit, August Pi i Sunyer Biomedical Research Institute (IDIBAPS), 08036 Barcelona, Spain; 14NUPROAS Handelsbolag, Nackă, Sweden; NUPROAS HB, Apartado de Correos 93, 17242 Girona, Spain

**Keywords:** functional virgin olive oil, olive oil phenolic compounds, thyme phenolic compounds, cholesterol efflux, HDL cholesterol, transcriptomics

## Abstract

The consumption of antioxidant-rich foods such as virgin olive oil (VOO) promotes high-density lipoprotein (HDL) anti-atherogenic capacities. Intake of functional VOOs (enriched with olive/thyme phenolic compounds (PCs)) also improves HDL functions, but the gene expression changes behind these benefits are not fully understood. Our aim was to determine whether these functional VOOs could enhance the expression of cholesterol efflux-related genes. In a randomized, double-blind, crossover, controlled trial, 22 hypercholesterolemic subjects ingested for three weeks 25 mL/day of: (1) a functional VOO enriched with olive oil PCs (500 mg/kg); (2) a functional VOO enriched with olive oil (250 mg/kg) and thyme PCs (250 mg/kg; FVOOT), and; (3) a natural VOO (olive oil PCs: 80 mg/kg, control intervention). We assessed whether these interventions improved the expression of cholesterol efflux-related genes in peripheral blood mononuclear cells by quantitative reverse-transcription polymerase chain reactions. The FVOOT intervention upregulated the expression of *CYP27A1* (*p* = 0.041 and *p* = 0.053, versus baseline and the control intervention, respectively), *CAV1* (*p* = 0.070, versus the control intervention), and *LXRβ*, *RXRα*, and *PPARβ/δ* (*p* = 0.005, *p* = 0.005, and *p* = 0.038, respectively, relative to the baseline). The consumption of a functional VOO enriched with olive oil and thyme PCs enhanced the expression of key cholesterol efflux regulators, such as CYP27A1 and nuclear receptor-related genes.

## 1. Introduction

Virgin olive oil (VOO) consumption protects against the development of cardiovascular diseases due to its richness in phenolic compounds (PCs) and other bioactive compounds such as monounsaturated fatty acids [1,2,3,4]. Among its beneficial mechanisms, the intake of olive oil PCs has been shown to promote major high-density lipoprotein (HDL) function, particularly cholesterol efflux capacity (HDL’s ability to pick up cholesterol from macrophages), as well as other improvements in HDL quality characteristics [5]. Going one step further, the consumption of enriched or functional VOOs, supplemented with olive oil PCs or other complementary PCs such as those from thyme, has also been shown to improve HDL functional traits and lipid profile [6,7,8,9,10], as well as other molecular mechanisms potentially involved in the development of cardiovascular diseases [11,12]. On the one hand, olive oil PCs are essentially monophenols such as hydroxytyrosol and tyrosol, whose effect on cardiovascular protection [13] (mainly based on their capacity to neutralize pro-oxidant substances) may be complemented with that from more complex PCs such as flavonoids (known to also be able to directly modulate pro-inflammatory pathways such as those involving phosphoinositol-3-kinase/protein kinase B, protein kinase C, and mitogen-activated protein kinases [14]). Beyond the antioxidant and anti-inflammatory effect attributed to olive oil PCs [15,16], their capacity to modulate gene expression is also essential to explain their health effects and those of the dietary patterns in which they are present, such as the Mediterranean diet [17,18,19]. The benefits of this healthy dietary pattern could be partially attributed to the protective capacities of VOO bioactive compounds on atherosclerosis and neoplastic processes [20]. On the other hand, thyme bioactive compounds such as PCs and essential oils have also been shown to be able to promote HDL cholesterol levels and to decrease low-density lipoprotein cholesterol concentrations in diverse research models [21]. 

In relation to HDL functionality, the acute consumption of a VOO enriched with its own PCs improved the expression of cholesterol efflux-related genes in pre-/hypertensive subjects in a previous trial [22]. However, the effects of the sustained consumption of functional VOOs on the gene expression related to cholesterol efflux and cholesterol metabolism regulation pathways have not been assessed to date. Thus, the aim of the present study was to assess whether a sustained consumption of functional VOOs (rich in complementary PCs from VOO or thyme) also promoted the expression of this gene pathway, in order to contribute to explaining their potential benefits on HDL function and lipid profile.

## 2. Materials and Methods 

### 2.1. Olive Oil Preparation

Two phenol-enriched VOOs (final PC concentration: 500 ppm) were prepared using as matrix a low-phenolic-content VOO (80 ppm), which also served as the control. In particular, the characteristics of the functional oils were: (1) a functional VOO enriched with an extra quantity of olive oil PCs (500 ppm) (FVOO); and (2) a functional VOO enriched with olive oil (250 ppm) and thyme PCs (250 ppm) (final PC concentration: 500 ppm) (FVOOT). Thyme PCs came from an aqueous thyme extract, not from an essential oil. The procedure to obtain the phenolic extracts and the enriched oils has been previously described [23]. 

### 2.2. Study Design and Biological Samples

Thirty-three hypercholesterolemic volunteers (total cholesterol > 200 mg/dL) participated in a randomized, double-blind, crossover, controlled trial in which we compared the effect of the three oils previously described (FVOO, FVOOT, and the control VOO). The intervention consisted of three-week periods in which the volunteers consumed 25 mL/day of raw VOO distributed along meals, preceded by two-week wash-out periods with an olive oil very poor in PCs. All subjects gave their informed consent for inclusion before they participated in the study. The study was conducted in accordance with the Declaration of Helsinki, and the protocol was approved by the local Research and Ethics Committees and registered with the International Standard Randomized Controlled Trial Number ISRCTN77500181 [6]. 

In the course of the trial, we collected blood samples at fasting before and after each treatment, in order to isolate peripheral blood mononuclear cells (which were conserved at −80 °C for further analysis). We also collected EDTA plasma samples to determine glucose, total cholesterol, and triglyceride concentrations using standard enzymatic automated methods in an ABX Pentra-400 autoanalyzer (ABX-Horiba Diagnostics, Montpellier, France), HDL cholesterol levels by an accelerator selective detergent method (ABX-Horiba Diagnostics), and computed LDL cholesterol by the Friedewald equation whenever triglycerides were <300 mg/dL. In this work, a sub-sample of 22 volunteers was analyzed on the basis of sample availability and quality. Further details of the clinical trial and the preparation of the functional oils have been thoroughly described in previous publications [6].

### 2.3. Gene Expression Analyses

First, we isolated total RNA from peripheral blood mononuclear cells from blood samples collected from volunteers before and after the three VOO interventions by RNAeasy microkits (Qiagen, Hilden, Germany), quantified total RNA, and checked RNA purity and integrity in a Nanoquant Bioanalyzer (Tecan Ltd., Männedorf, Switzerland). We then converted RNA into complementary DNA using High Capacity cDNA Reverse Transcription kits (Applied Biosystems‒Life Technologies, Carlsbad, CA, USA). Finally, we quantified gene expression by the quantitative retrotranscriptase real-time polymerase chain reaction (qRT-PCR) technique, using TaqMan Low Density microfluidic cards (Applied Biosystems‒Life Technologies). We selected our candidate genes based on their relationship with cholesterol efflux (cholesterol transporters and efflux transcriptional regulators). We analyzed the data using the Sequence Detection System software 2.4 (Applied Biosystems‒Life Technologies) and calculated the changes in gene expression by the relative quantification method (applying the 2^−ΔΔCt^ formula) [22].

### 2.4. Sample Size

A sample size of 22 participants allowed ≥80% power to detect significant inter-group differences of 0.26 units in the relative quantification of ATP-binding cassette, subfamily A1 (*ABCA1*) expression, considering a two-sided type I error of 0.05, a dropout rate of 10%, and the standard deviation of the differences of *ABCA1* gene expression after an analogous intervention with a PC-rich VOO [22].

### 2.5. Statistical Analysis

We evaluated the normality of continuous variables by normal probability plots. We discarded differences in baseline parameters between the eligible volunteers and the whole study population with chi-squared tests, *t*-tests, and Mann‒Whitney U tests (in categorical, normally, and non-normally distributed continuous variables, respectively). We assessed the effects of each intervention relative to baseline by paired *t*-tests, and the inter-treatment differences in mixed models adjusted for age and sex, in which we also studied period-by-treatment interactions to discard possible carry-over effects. To study whether the differences in gene expression variables after the oil interventions were related among them, we assessed the associations among post-intervention changes by Spearman’s correlation analyses. We considered significant any *p*-value < 0.05 and performed all statistical analyses in R Software (Vienna, Austria) v3.1.0 [24].

## 3. Results

### 3.1. Study Participants

The 22 eligible volunteers presented no differences in their baseline characteristics relative to the whole trial population (Table 1).

### 3.2. Gene Expression Analyses

FVOOT intake upregulated significantly the expression of cytochrome P450, subunit 27A1 (*CYP27A1*; *p* = 0.041), liver X receptor beta (*LXRβ*; *p* = 0.005), peroxisome-proliferator activated receptor beta/delta (*PPARβ/δ*; *p* = 0.038), and retinoid X receptor alpha (*RXRα*; *p* = 0.005), and the expression of *ABCA1* in a marginally significant way (*p* = 0.099), relative to baseline in all cases. Upregulation of caveolin-1 (*CAV-1*), *CYP27A1*, and *LXRβ* was marginally significant relative to the VOO control intervention (*p* = 0.070, *p* = 0.053, and *p* = 0.070, respectively). Finally, VOO intake downregulated *RXRβ* expression relative to baseline (*p* = 0.025) (Table 2). 

*CAV1*, *CYP27A1*, *LXRβ*, *PPARβ/δ*, and *RXRα* upregulation after the FVOOT intervention were intimately inter-correlated and associated with increments in *ABCA1* expression (*r* > 0.55, *p* < 0.0042 ‒adjusted *p*-value for 12 comparisons, according to the Bonferroni method‒) (Figure 1).

## 4. Discussion

In the present work, we report that the consumption of a functional VOO, enriched with olive oil and thyme PCs, enhanced the expression of key cholesterol efflux genes.

The ability of HDL to promote cholesterol efflux from macrophages is the most studied HDL function and reflects HDL anti-atherogenic role better than HDL cholesterol levels [25]. Lipid-poor and small HDLs mediate cholesterol efflux by binding to ABCA1 (in a cellular system intimately related to caveolae/CAV1 gene expression in immune cells such as macrophages) [26] and large HDLs exert this process via ATP-binding cassette G1 and scavenger receptor B1. Expression of these cholesterol transporters is mainly regulated by LXRs and PPARs, key transcription factors in the overall regulation of lipid metabolism [25,26]. In our study, consumption of the FVOOT functional oil promoted *LXR* gene expression and activation through two hypothetical pathways (Scheme 1): (1) the upregulation of *CYP27A1* (a monooxygenase member of the cytochrome P450 superfamily), enzyme responsible for the transformation of cholesterol into oxidized sterols, essential activator ligands of LXR; and (2) the upregulation of *RXRα*, whose correspondent protein is necessary to build the LXR-RXR heterodimer (the active complex responsible for LXR nuclear functions) [27]. A quantitative proteomic approach performed in our study had already reported an increase in the levels of several proteins related to the LXR/RXR axis after the functional and the control VOO interventions [28]. Activation of LXR function has been consistently linked to increments in the expression of cholesterol transporters involved in the efflux phenomenon (such as ABCA1 and ABCG1) [25] and of *CAV1* [26]. In addition, FVOOT intake upregulated *PPARβ/δ*, another transcription factor known to induce the expression of *ABCA1* [29] and *CAV1* [30]. Increases in the expression of *RXR* may be additionally related to a greater activation of this PPAR (its active form is also a heterodimer with RXR), which, in turn, is also known to promote LXR function [29].

Olive oil and thyme PCs are not the only known to be able to modulate cholesterol metabolism-related gene expression pathways. Other bioactive dietary compounds such as curcumin and 6-dihydroparadol from ginger, and coenzyme Q10 have been shown to promote cholesterol efflux-related gene expression in previous in vitro or in vivo studies [31,32]. Furthermore, nutritional supplements of green tea catechins have also been associated with a promotion of the *RXR* expression in animal models [33]. Finally, coenzyme Q10 treatment promoted ABCG1-mediated macrophage cholesterol efflux in a randomized trial in healthy volunteers [34]. Upregulation of some of these genes (*PPAR*-related, *ABCA1*, and *SRB1*) has also been reported after an acute consumption of a PC-rich VOO in pre-/hypertensive subjects [22]. However, in the present study, the FVOOT intervention promoted the expression of cholesterol-efflux regulatory genes only versus their baseline, although marginally significant increases in their expression were also observed relative to the control intervention. Differences among studies can be explained by the extremely high content in PCs administered in the previous work [22], the study design (acute use versus mid-term sustained consumption), and the pathophysiology of the participants (hypertensive versus hypercholesterolemic). 

The greatest strength of this study is its randomized, crossover, double-blind, controlled design, which confers strong inter-individual variability control. However, it also presents limitations: the observed effects were expectedly modest, since our trial was based on oil doses that could be achievable via a regular diet, and the control intervention was a VOO (with a considerable amount of olive oil PCs, 80 ppm), which makes it more difficult to observe inter-group differences.

## 5. Conclusions

A functional VOO, enriched with olive oil and thyme PCs (250 mg/kg of each), promoted the expression of several genes regulating HDL cholesterol efflux capacity (*CAV1*, *CYP27A1*, *LXRβ, PPARβ/δ*, *RXRα*). Such an improvement may contribute to explaining the beneficial effect of this functional food product on HDL functional capacities in hypercholesterolemic patients. These results contribute to explaining the beneficial effects of these functional oils towards a more cardioprotective lipid profile.

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
