# Peer review of "A Functional Virgin Olive Oil Enriched with Olive Oil and Thyme Phenolic Compounds Improves the Expression of Cholesterol Efflux-Related Genes: A Randomized, Crossover, Controlled Trial"

_nutrients, 2019, doi:10.3390/nu11081732_

Round 1

Reviewer 1 Report

The authors answered all my questions adequatelly.

This manuscript is a resubmission of an earlier submission. The following is a list of the peer review reports and author responses from that submission.

Round 1

Reviewer 1 Report

The present manuscript entitled "A functional virgin olive oil enriched with olive and thyme phenolic compounds improves the expression of efflux-related genes: a randomized, cross-over, controlled trial" evaluates the effect of different olive oils enriched in phenolic compounds on the expression of genes related to cholesterol efflux in PBMCs, finding that oil enriched in thyme phenolic compounds has the most effect in relation to this. However, there are certain aspects to improve:

Major revisions:

- In Material and Methods it has been indicated that the intervention consisted of 3-week periods in which the volunteers consumed 25 mL/day of raw VOO distributed along meals, preceded by 2-week wash-out periods with a common olive oil. What the term exactly refers to common olive oil?

-A spearman’s correlation analysis of the different genes was performed, but I think that the reason for its use or  the relevance of its result. Please,it possible to clarify this issue inMaterial and methods and discussion.

-In the present study, the use of VOO only modify RXRB gene expression which did not occur with other oils, If the VOOis the basis for the other products, how it this possible? Please disccus this issue.

-According to the differences in gene expression, it seems that compounds from thymus result more relevant for health benefits but Introduction is mainly focused on virgin olive oil and related compounds properties. I think that that use of thymus or thymyus-derived compounds must be justified in Introduction

-In Introducction it has been mentioned that the acute consumption of a VOO enriched with its own PCs improved the expression of cholesterol efflux-related genes in pre-/hypertensive subjects in a previous trial. However, the results of this study suggest that this effect would not be due to change in gene expression, it is possible to discuss this fact?

Minor revisions:

-In Introdution authors said that "Virgin olive oil (VOO) consumption protects against the development of cardiovascular diseases due to its richness in phenolic compounds (PCs) and other bioactive compounds [1–3]", but it has been reported that n-9 fatty acids could be also responsible for this effects (e.g. doi:10.1093/ajcn/70.6.1009; doi: 10.3390/nu4121989). Could be this fact mentioned?

-In Introduction authors mentioned that "Beyond the antioxidant and anti-inflammatory effect attributed to olive oil PCs [14,15], their capacity to modulate gene expression is also essential to explain their health effects and those of the dietary patterns in which they are present, such as the Mediterranean diet [16]". However, there is studies reporting differential expression of genes as consequence of the specifical consumption of virgin olive oil (e.g doi:10.1096/fj.09-148452; doi:10.1080/10408398.2018.1526165). I think that these should be included.

-Regarding HDL functions, the acute consumption of a VOO enriched with its own PCs improved
the expression of cholesterol efflux-related genes in pre-/hypertensive subjects in a previous trial [17]. Regarding HDL functions,?

- In Material and Methods it has been mentioned that "Along the trial, we collected blood samples at fasting before and after each treatment, in order to isolate peripheral blood mononuclear cells (which were conserved at -80oC for further analysis) as well as EDTA-plasma samples. Please

- In spite of the number of participants of the study in this work, a sub-sample of 22 volunteers was analyzed, why?

Author Response

Response to Reviewer 1 Comments

Point 1: In Material and Methods it has been indicated that the intervention consisted of 3-week periods in which the volunteers consumed 25 mL/day of raw VOO distributed along meals, preceded by 2-week wash-out periods with a common olive oil. What the term exactly refers to common olive oil?

Response 1: As stated in reference #24 (in which the study protocol is described), volunteers consumed an olive oil very poor in phenolic compounds. This olive oil is generally described as “common” in the literature and among consumers and comes from the mixture of refined (which has lost the vast majority of its antioxidants in the refining process) and virgin olive oil in a very small proportion. The very poor content in antioxidants of this olive oil makes it a suitable product to replace virgin olive oil between interventions. We agree with the reviewer that it is not clear that the term “common olive oil” refers to an olive oil of these characteristics and, therefore, we have substituted this expression for “olive oil very poor in phenolic compounds”. We will rephrase this aspect in the text (line 74).

Point 2: A Spearman’s correlation analysis of the different genes was performed, but I think that the reason for its use or the relevance of its result [is not clear]. Please, clarify this issue in Material and methods and Discussion.

Response 2: These analyses were performed to assess whether the changes we observed in gene expression after the FVOOT intervention were related among them. The interrelation between changes in gene expression makes their interconnection in a signaling cascade more plausible, as described in Scheme 1. This is now stated in lines 116-118.

Point 3: In the present study, the use of VOO only modified RXRB gene expression which did not occur with other oils. If the VOO is the basis for the other products, how it this possible? Please discuss this issue. 

Response 3: We hypothesize that higher levels of antioxidants or the combination of complementary phenolic compounds (in the particular case of the FVOOT intervention) may induce differential effects on gene expression. Indeed, the VOO is the control intervention of the study, and it is not particularly rich in phenolic compounds (80 ppm, versus 500 ppm in functional oils). Therefore, observing an effect associated to this oil and not linked to the consumption of the others could be expected.

In addition, there was a mistake in the discussion of RXR-related results that has been corrected (line 167, we indicated there was a promotion in the expression of RXRβ due to the FVOOT intervention but the upregulation happened for the RXRα gene).

Point 4: According to the differences in gene expression, it seems that compounds from thymus result more relevant for health benefits but Introduction is mainly focused on virgin olive oil and related compounds properties. I think that that use of thymus or thymus-derived compounds must be justified in Introduction.

Response 4: We totally agree with Reviewer #1 in this point. A description of the effects on thyme phenolic compounds and bioactive substances on lipid profile parameters is now available in the Introduction (lines 48-50) in order to clarify this aspect.

Point 5: In Introduction it has been mentioned that the acute consumption of a VOO enriched with its own PCs improved the expression of cholesterol efflux-related genes in pre-/hypertensive subjects in a previous trial. However, the results of this study suggest that this effect would not be due to change in gene expression. Is it possible to discuss this fact?

Response 5: In the text (lines 51-53) we indicate that “the acute consumption of a VOO enriched with its own phenolic compounds improved the expression of cholesterol efflux-related genes in pre-/hypertensive subjects in a previous trial”. Our present study also suggests that the consumption of a functional VOO is able to improve the expression of cholesterol efflux-associated genes. In both cases, this mechanism is one of the possible hypothesis to explain the beneficial effects of these dietary interventions on lipid profile. We have added an extra sentence after the initial expression in order to clarify these aspects (lines 53-55). Differences among studies can be explained by the extremely high content in PCs administered in the first work, the study design (acute use versus mid-term sustained consumption), and the pathophysiology of the participants (hypertensive versus hypercholesterolemic), as is explained in the Discussion (lines 193-196).

Point 6: In Introduction authors said that "Virgin olive oil (VOO) consumption protects against the development of cardiovascular diseases due to its richness in phenolic compounds (PCs) and other bioactive compounds [1–3]", but it has been reported that n-9 fatty acids could be also responsible for this effects (e.g. doi:10.1093/ajcn/70.6.1009; doi: 10.3390/nu4121989 ). Could be this fact mentioned?

Response 6: We have added a mention to monounsaturated fatty acids (lines 31-32) and an extra reference to the text (the second one suggested by the reviewer).

Point 7: In Introduction authors mentioned that "Beyond the antioxidant and anti-inflammatory effect attributed to olive oil PCs [14,15], their capacity to modulate gene expression is also essential to explain their health effects and those of the dietary patterns in which they are present, such as the Mediterranean diet [16]". However, there is studies reporting differential expression of genes as consequence of the specifical consumption of virgin olive oil (e.g doi:10.1096/fj.09-148452;doi:10.1080/10408398.2018.1526165). I think that these should be included. 

Response 7: Both references have been included in the text.

Point 8: Regarding HDL functions, the acute consumption of a VOO enriched with its own PCs improved the expression of cholesterol efflux-related genes in pre-/hypertensive subjects in a previous trial [17]. “Regarding HDL functions”?

Response 8: We have reworded this expression (line 51).

Point 9: In Material and Methods, it has been mentioned that "Along the trial, we collected blood samples at fasting before and after each treatment, in order to isolate peripheral blood mononuclear cells (which were conserved at -80oC for further analysis) as well as EDTA-plasma samples. Please […].

Response 9: We understand that the reviewer wants us to explain why we collect EDTA plasma samples. It is now stated in line 81.

Point 10: In spite of the number of participants of the study in this work, a sub-sample of 22 volunteers was analyzed, why?

Response 10: We could not use the samples of the whole VOHF study population due to sample availability issues (the extraction of peripheral blood mononuclear cells from whole blood is a critical step and can be linked to sample loss and decreases in the quality of the specimens). It is stated in the text in line 86.

Reviewer 2 Report

The reviewed paper entitled:

A functional virgin olive oil enriched with olive and thyme phenolic compounds improves the expression of cholesterol efflux-related genes: a randomized, cross-over, controlled trial

The material presented to my person for review clearly shows that the use of a functional VOO enriched with PC from olives and thyme has increased the expression of key regulators of cholesterol outflow, such as the CYP27A1 gene. The functional VOO, enriched with olive and thyme PC (250 mg / kg), promoted the expression of several genes regulating the ability of HDL cholesterol (CAV1, CYP27A1, LXRβ, PPARβ / δ, RXRα) outflow. The article is written succinctly and clearly, thus the key results are well presented and can be traced easily. Although in my opinion the conclusions are too general thus giving the impression of unfinished studies.

The paper seems to be acceptable (especially in this journal section) but, in my opinion, it requires some modifications. Additionally, several questions should be answered by the authors in detail, as many important issues are described too superficially:

Line 13: Regarding thyme and its compounds, it      should not be used by pregnant women. Thyme lowers blood sugar. Thyme oil      is considered toxic, so it should not be used internally.

Line 29-51: It is worth describing more broadly      that a diet rich in olive oil has a very high inhibitory effect on the      development of atherosclerosis. It was found that biologically active      compounds present in olive oil have a large and effective protective      effect against the formation and development of cancer cells. Studies in Spain      showed lower overall mortality among older people which diet is enriched      of olive oil.

Line 194-197: The conclusions are too general and      give the impression of unfinished studies; this section should be better      developed.

In conclusion, the paper seems to be acceptable but requires some revisions. The whole layout and neatness of the paper do not leave too much objections, as it is prepared very carefully, but the quality of the discussion requires several amendments. Please answer all my questions and comments and attach the manuscript with marked changes. The objections presented by me do not undermine the quality of the paper, which will be support in the further publishing process, certainly after careful consideration of my comments.

Author Response

Response to Reviewer 2 Comments

Point 1: Line 13: Regarding thyme and its compounds, it should not be used by pregnant women. Thyme lowers blood sugar. Thyme oil is considered toxic, so it should not be used internally.

Response 1: As indicated by the reviewer, an essential thyme oil could be irritant to some patients due to its high content of volatile monoterpenes. However, the functional olive oil enriched with phenolic compounds from thyme and olive oil (FVOOT) used in this study was not prepared by supplementing an essential thyme oil, it was elaborated by adding a thyme aqueous extract to a basic VOO. In addition, the doses of thyme phenols attained by this addition lead to those achievable by a normal thyme intake. The preparation of the functional olive oils of our trial has been described in a previous reference (Rubió L, J Agric Food Chem, 2012, reference #22 in the text). However, it is true that the functional oil preparation could be clarified in the text. Thus, we have reworded this description in lines 65-66.

Point 2: Line 29-51: It is worth describing more broadly that a diet rich in olive oil has a very high inhibitory effect on the development of atherosclerosis. It was found that biologically active compounds present in olive oil have a large and effective protective effect against the formation and development of cancer cells. Studies in Spain showed lower overall mortality among older people which diet is enriched of olive oil.

Response 2: Considering that we are discussing potential mechanisms to explain the cardioprotective benefits of VOO-related products, we have included an extra sentence about the healthy effects of VOO-rich dietary patterns such as the Mediterranean diet in the Introduction (lines 46-48, extra reference #20).

Point 3: Line 194-197: The conclusions are too general and give the impression of unfinished studies; this section should be better developed.

Response 3: We have added a stronger conclusion in lines 206-208 following the recommendations of Reviewer 2.

Point 4: In conclusion, the paper seems to be acceptable but requires some revisions. The whole layout and neatness of the paper do not leave too much objections, as it is prepared very carefully, but the quality of the discussion requires several amendments. Please answer all my questions and comments and attach the manuscript with marked changes. The objections presented by me do not undermine the quality of the paper, which will be support in the further publishing process, certainly after careful consideration of my comments.

Response 4: We would like to thank Reviewer #2 for his/her kind comments about our work and hope we have addressed correctly his/her criticisms.
